# Seafarers’ Perception and Attitudes towards Noise Emission on Board Ships

**DOI:** 10.3390/ijerph18126671

**Published:** 2021-06-21

**Authors:** Luka Vukić, Vice Mihanović, Luca Fredianelli, Veljko Plazibat

**Affiliations:** 1Department for Maritime Management Technologies, Faculty of Maritime Studies, University of Split, Ruđera Boškovića 37, 21000 Split, Croatia; veljko.plazibat@pfst.hr; 2Port Authority Split, Gat Svetog Duje 1, 21000 Split, Croatia; vice.mihanovic@gmail.com; 3Physics Department, University of Pisa, Largo Bruno Pontecorvo 3, 56127 Pisa, Italy; fredianelli@df.unipi.it

**Keywords:** seafarers, acoustic pollution, noise onboard ship, health impact, environmental pollution, noise survey

## Abstract

Noise has long been neglected as an environmental pollutant and impairment health factor in maritime transport. Recently, acoustic pollution indicates the highest growth in transport external cost unit values. In 2020, questionnaires were submitted to seafarers to examine their noise exposure and perception on board and attitudes towards noise abatement measures. Responses of 189 participants were processed using descriptive statistics and Likert scale valuation, while their consistency was tested with indirect indicators using linear regression and correlation test. Results show that more than 40% of respondents do not consider noise as a significant environmental problem. The negative perception among respondents with ≥10 years of work experience was much lower (23.53%). Most are aware of the onboard noise harmful effects that can influence their health. Despite that, they use personal protection equipment only sometimes. A higher positive perception was recorded in groups of respondents with a university degree (90%), work experience longer than ten years (82.35%), and monthly income higher than 4000 € (70%). Respondents are not strongly motivated to participate in funding noise mitigation measures, and such a viewpoint is not related to their monthly incomes. The low awareness and motivation regarding acoustic pollution generally shown by the surveyed seafarers should be watched as a threat by the company managers. Better education and awareness are likely to be crucial to change the current state of affairs.

## 1. Introduction

The negative impact of transport on the environment and human health is usually expressed through external costs, where the noise cost has recently become a significant source of damage. These costs are not covered by the stakeholders of the logistics transport chain but are a burden to society. External cost is expressed as a price per unit of harmful transport product (e.g., decibels (dB) for noise). Based on the recent data on the external costs in transport retrieved from relevant literature [1,2,3], the noise external costs unit prices have increased more than 3.5 times in the last 12 years, an increase not recorded in any other external cost component in the sector. Reasons are changes in perception of noise pollution, modified regulations, insufficient and expensive protection measures, and stricter valorization due to recent findings of the noise impact on health. Recently, noise costs have become a significant factor in the transport impact on human health and the environment, accounting for almost 7% of total external transport costs in the European Union (EU) [3].

The World Health Organization (WHO) has recognized noise pollution as not only an environmental nuisance but a threat that can damage health and reduce the nearby property value [4]. More than 20% of EU residents have been exposing to an excessive noise level [5]. Prolonged exposure to noise levels above 55 dB(A) can be detrimental to health, while levels above 65 dB(A) should not be tolerated [6] over the long term. The health effect of noise starts from the “indirect” ones, such as annoyance (nuisance), sleep disturbance, stress, anxiety occurring at lower levels of exposure, and “direct effects” when the exposure exceeds 85 dB(A). Direct effects include tinnitus, cognitive impairment in children, ischemic heart disease, and hypertension [7]. Also, for these reasons, noise has been recognized as one of the main reasons for the reduced life quality in urban and country areas [8,9].

The transportation sector is the principal cause of environmental noise, where road contributes to 65%, air to 20%, and railway to 15% of the overall level of noise impact in the environment [8]. Maritime and inland waterways transports have a reduced significance [3] with the consequence that few studies have been published in the scientific literature. However, ship noise onboard can endanger seafarers and passengers, while underwater and airborne emitted ship noise can affect port areas and coastal residents, even the fauna on maritime routes [10]. Based on the research of [11,12,13,14], the principal source of noise on board can be assumed to be the engine room, where the highest levels of intensity can be found. On most ships, noise levels over 100 dB(A) are present, reach the levels of 110 dB(A) in the noisier area and decrease depending on the location on board. Permanent and simultaneous exposure to noise, vibration, and heat on ships contributes significantly to developing anxiety in seafarers [11]. Noise exposure onboard increases mobility during sleep by 12%, and conjoined with other agents like caffeine and nicotine, may cause shallow sleep [15]. A better rest improves health and safety, which indirectly reduces the frequency of onboard accidents and improves productivity [16]. There is still debate about the relationship between ship noise and arterial hypertension occurrence in seafarers [17]. Hearing loss is a leading occupational disease, and seafarers working in an engine room on a ship are particularly at risk [18]. The Norwegian Centre for Maritime Medicine reviewed noise levels on board and their influence on seafarers [19]. Nastasi et al. [20] point out that noise has only recently been taken into account in the port sustainability assessment. Exposure of citizens to the noise in port areas has also been underestimated [21]. In the port of Livorno, e.g., during arriving and departing ships, the noise increases by 6–10 dB above the existing background noise [22]. Witte [23] states that the mitigating measures of ship noise at berth, like shore power connection, can drastically improve air quality but not reduce noise emission proportionally.

In 2012, the International Maritime Organization (IMO) adopted the Convention for the Safety of Life at Sea (SOLAS) with a requirement for noise reduction, both by adequate solutions in ship construction and personal protection equipment for seafarers following The Code on noise levels on board ships [24]. The Code has been developed to provide international standards for protection against noise and tools to promote “hearing saving” environment onboard ships. Unfortunately, not enough public awareness of the harmfulness of noise on ships and in ports [25] has been raised since then. Raising awareness and education about the harmful effects of noise is crucial, and such initiatives come from all over [26]. Despite regulations, the intensity of noise on ships often exceeds the permissible values determined by Directive 2003/10/EC [12,13,27]. There is also relatively little interest in the scientific community, and papers on noise as working environment and barrier to development are not frequent.

When exposed to environmental noise levels between 50 and 75 dB(A), noise experience and acceptance vary on individual. Also, the noise tolerance threshold is determined independently, as one can tolerate higher noise intensities while another cannot tolerate noises below 50 dB regardless of education on the detrimental effects of noise. This aspect led scientists to introduce the term noise sensitivity. It is a measuring unit of non-auditory influence of the environmental noise, which is individually different at the same intensity noise exposure [28]. Some other adverse factors have collateral effects on noise perception, such as meteorological conditions or, in general, changing conditions at the site of perception. Therefore, valorization based on statement, impression, attitude, and opinion is imprecise and uncertain, and the possibility of objectifying disorders is limited.

The present paper aims to determine the seafarers’ noise pollution perception on board and evaluate their attitudes towards noise exposure. The aim is reached using a structured questionnaire based on collecting general noise perception data on environment and health, as well as noise perception on board and in place of residence. Encouraged by the current trend and sudden increase in the external noise costs, the research would contribute to the topic’s actuality. Noise cost marginalization in maritime transport refers only to the low capital share and does not to the real significance of noise pollution. The research also wishes to contribute and drive the education and raising seafarers’ awareness of the noise harmfulness on board. Awareness level about the harmfulness of noise in people who are professionally exposed to it and therefore may suffer health consequences is a good indicator of how much significance is attached to noise as an environmental pollutant.

## 2. Materials and Methods

A structured questionnaire (Appendix A) for seafarers was composed about the perception and intensity of noise pollution in general and onboard ships. Noise analysis is combined with the top-down approach through the willingness to pay value (WTP), and alternatively, willingness to accept (WTA), multiplied by the number of noise-exposed persons to obtain average or total external noise costs [7,29]. Thus, the noise valorization is identified with the people’s motivation in how much they are willing to spend for implementing the measures that will reduce the noise and, alternatively, how much compensation they claim for noise tolerance. Awareness of the harmful effects of noise is of great importance for conducting such a survey. When awareness of the noise exposure detrimental effects is not sufficient, a credible response can be obtained indirectly using a hedonic pricing method (HP). The method enables estimating one’s attitudes towards noise pollution over his/her opinion on whether and to what extent noise affects own real estate prices and rental prices [7]. The present paper examined the seafarers’ willingness to participate in financing noise abatement (WTP) as a good indicator of what extent an individual attaches importance to the topic. The respondents’ objectivity was tested by questions about the need for a salary supplement due to noise exposure (WTA), perception of noise in own household, and noise impact on the own apartment value (HP). For the simple estimation of the noise intensity to which they are exposed, the respondents could use a decibel level comparison table attached in the questionnaire and choose the option. To some questions, respondents had to answer using the Likert scale. Data were processed using descriptive statistics. The correlation test (CORREL) and linear regression (LR) were used to determine the dependency between the size of monthly income (MI) and WTP as well as the requirement for a salary supplement due to noise impact (WTA) and WTP. The possible WTA and WTP values correlation with the estimations on the own apartment values loss due to noise (HP) were also determined. All calculations were made in spreadsheets. The methodological concept applied as sketched in Figure 1 aimed to objectify the consistency of the responses.

Expectations from respondents, who were occupationally exposed to noise pollution and aware of the harmful effects of noise on the environment and human health, are as follows: that those with higher monthly income will contribute more for noise mitigation (WTP/MI); that those who seek higher compensation for occupational noise pollution will contribute more for noise mitigation (WTP/WTA); that those who contribute more for noise mitigation also estimate the greater loss in value of their property due to noise (HP/WTP); that those who seek higher compensation for occupational noise exposure simultaneously estimate the corresponding loss in value of their property due to noise (HP/WTA).

In order to exclude subjectivity in the choice of answers, the F-test was used to examine the response dispersion differences to the noise perception at work and in their household. The same was examined in the groups of participants who indicated a possible leaving from the ship, respectively, changing the housing location due to noise exposure. All calculations were made in MS Excel.

The research was conducted from February to June 2020 at the Faculty of Maritime Studies in Split, Croatia. All respondents were participants in the course of additional education of seafarers (which is not related to a topic of noise). All respondents were Croatian citizens.

## 3. Results

In 2020, the questionnaire was applied to 189 seafarers with an average age of 35 years (27–52 years) and an average work experience of 11.5 years (4–29 years) with a median of 10 years (y). An average income was 3250 € a month (1000–5000 €). They work on merchant and passenger ships, being on board continuously for at least two months, followed by a month’s rest on land. There were 171 male and 18 female seafarers in the research. The perception of respondents is shown in Table 1.

The research results show that 41.27% of respondents do not consider noise pollution a significant environmental problem. Concerning education, almost the same percentage of the above perception was recorded among the respondents with secondary education (39.62%). It unexpectedly increased to 50% among those with higher education levels. Dispersion of respondents by the work experience in years is reported in Figure 2.

The median of 10 years was the criteria for creating comparative groups, a group <10 y, *n* = 87, and a group ≥10 y, *n* = 92. The variance examined with the two tail F tests shows statistically significant difference (*p* = 3.09 × 10^−33^, α = 0.05). The negative perception among respondents with ≥10 years of work experience was much lower (23.53%) compared to respondents with <10 years of work experience (47.83%). Monthly income does not affect the perception of noise pollution. The statement that air pollution in maritime transport is a bigger problem than noise support 93.44% of the seafarers surveyed.

More than 50% of respondents are aware of the harmful effects of noise on health, more than 25% are aware of this at least partly, and 19% of respondents deny them. A higher positive perception was recorded in groups of respondents with a university degree (90%), work experience longer than ten years (82.35%), and monthly income higher than 4000 € (70%).

On a Likert scale, ranging from 1 to 5, respondents rated noise exposure on board as 3.85 (1 = does not interfere at all, 2 = interferes very little, 3 = little, 4 = much, 5 = very much), and equally during working hours (3.11) and rest periods (3.15). According to the attached intensity table, the estimated noise intensity during working hours is supposed at a range of 80–85 dB, and during rest hours at a range of 50–55 dB. The share of seafarers willing to provide salary supplement due to noise exposure was 5.75%. About 13.33% of respondents considered leaving the ship due to noise. On a Likert scale range from 1 to 3, the noise protection equipment use was at 2.37 (1 = never, 2 = sometimes, 3 = always). Vibration exposure on the same scale was rated with 2.22.

The surveyed seafarers indicated a willingness to pay an average of 65 € per year for noise mitigation. The dependence of the size of payments declared for noise mitigation on monthly incomes was examined by linear regression, as reported in Figure 3.

The dependence between the given parameters was not determined (R^2^ = 0.00006). The correlation test obtained value, r = 0.0075, confirms the absence of any relationship.

Furthermore, the dependence of the size of the payment declared for noise abatement on the request size for salary supplement due to noise was examined by linear regression as reported in Figure 4. Even this resulted not to be determined (R^2^ = 0.018). The correlation coefficient r = 0.13398 indicates a very weak positive correlation.

On a Likert scale range from 1 to 5, respondents rated the perception of noise in their households with 2.27 (1 = does not interfere at all, 2 = interferes very little, 3 = little, 4 = much, 5 = very much), mostly at night (2.05 at a Likert scale range from 1 to 4 (1 = does not interfere at all, 2 = interferes at night, 3 = during the day, 4 = day and night). According to the attached table, they estimated the intensity of the household noise in the range between 50–55 dB by day and 35–40 dB at night. The surveyed seafarers believe that noise affects the value of the apartment by an average of 9.77%. Only 11.29% of respondents considered moving from their residence due to noise.

The variance differences in response groups on noise perception at the respondents’ workplace and their homes were examined using the *F* test. The same procedure was applied to groups who declared intention to leave the workplace and move from their apartments due to noise, respectively. There were no statistically significant differences in variance among groups (*p* = 0.2910, one tail; *p* = 0.1699, one tail). The correlation test result, r = 0.1961, shows a very weak positive correlation between the last two groups.

The dependence of attitudes about the noise impact on own apartment value on those about the salary supplement request due to noise at the workplace was examined by linear regression, as reported in Figure 5. The low coefficient of determination (R^2^ = 0.0546) indicates a minimal degree of dependence between the two groups of responses. The correlation value determined by the correlation test, r = 0.23363, shows a very weak positive correlation between the examined groups.

The same tests were used to find the dependence of attitudes towards the noise impact on the own apartment value on attitudes towards a voluntary contribution for noise abatement, as reported in Figure 6. The low coefficient of determination R^2^ = 0.0095 and a correlation coefficient r = 0.0973 are found, indicating the absence of dependence and correlation between the settings.

## 4. Discussion

The submitted questionnaires showed that almost half of the surveyed seafarers, in general, do not perceive onboard noise as a significant environmental problem in maritime transport, even if they are aware that prolonged noise exposure can have consequences for their health. According to the European Environmental Agency [30], this phenomenon happens to other people too. Subjective responses to noise depend not only on exposure levels but also on personality traits, expectations, and situational factors [31,32]. The results showed a noise harmfulness better perception in seafarers with more work experience, and noise health impact perception was also better in those with higher education and income. Choosing appropriate values, surveyed seafarers estimated their noise exposure level on board by the intensity that can damage their health and compromise their rest hours. The estimated average noise intensity during working hours was at almost 85 dB. This value follows findings obtained by Oldenburg et al. [11] and measured by Mansi et al. [14]. They are, obviously, insufficiently protected as they use noise protection agents only occasionally. Despite the actual situation, seafarers are not ready to invest significant funds in noise mitigation, not even when it comes to their health. The amount of the declared financial contribution does not depend on the monthly income or whether they receive a monthly allowance for working in noise. This attitude objectifies the level of perception of noise pollution. The perception of noise in own apartment is consistent with the perception in the workplace. In general, respondents do not want to leave the workplace due to noise nor consider moving out of the apartment. Their attitudes to the need for noise reduction are inconsistent. The absence of any dependence of the amount of contribution for noise reduction on control indicators and control indicators on each other indicates other motives for such selection concerning the adopted attitudes about noise hazards. A similar conclusion has been published by Picu et al. [33]. Noise pollution has not sufficiently become aware among seafarers even though they are directly exposed to it in the workplace, contrary to air pollution, which they are more exposed to globally than locally. Insufficient education is probably the main reason for the weak perception of noise pollution among seafarers. A low level of perception by seafarers with a university degree could present a confirmation of this thesis. The lack of knowledge was the main reason for the port authorities’ response to a special call for noise within the Interreg Maritime program [34].

The paper of Bernotaitė and Malinauskienė [35] found noise disturbance prevalence among seafarers of 15.6%, which is similar to the number of respondents in this study who considered leaving a ship due to noise (13.3%). The results show that noise pollution on board is not only temporary but permanent. Moreover, the research conducted by Szczepański and Otto [36] long ago found that noise levels during travel over and over exceed accepted norms, and reversible hearing impairment has been recorded after just one trip already.

Noise perception is an uncertain category. The estimated number of people exposed to noise is always lower than realistic. The number of exposed people who have disturbances due to noise exposure is uncertain as it is often a subjective assessment of an individual. Noise propagation from a single source is variable, while the spread from multiple sources is fraught with uncertainty. Noise protection measures can be primary, reducing noise at source and secondary such as noise propagation prevention, noise protection at home and workplace, economic measures, and regulations. They are individually very costly, and their effectiveness is generally low or uncertain [8]. However, Bowes et al. [37] showed that the costs of treatment and other compensation for hearing loss on navy ships are 15 times higher than investing in prevention programs, which offers, among other benefits, the possibility of significant savings.

## 5. Conclusions

Although increasingly supported by scientific evidence, the impact of noise on health has not yet been accepted as a real danger remaining underestimated without reaching full social awareness. Methods for external noise costs calculation remain subjective. The uncertainty of the noise nature and the limited motivation of the research community are reasons that little have been done to reduce noise in line with sustainable transport development. It is necessary to raise awareness of the damage caused by transport and its possible influence on the decision-making process in selecting the most appropriate transport mode. Education is crucial in raising awareness of noise detriment. The recent findings on the noise impact reveal greater exposure and more comprehensive health disorders than previously thought. This study contributes to raising awareness and the overall perception of noise pollution in maritime affairs, but with a small sample of seafarers, which cannot be considered representative, limits the results values. Within a surveyed period, seafarers underwent additional training, and their knowledge might be better than in the general population of seafarers. Furthermore, unlike the general population, this group is occupationally exposed to noise, and thus attitudes towards noise pollution are likely to be partly personally motivated. Limited perception and attitudes toward noise on board would probably be even more prominent by removing weaknesses from the research. Further research should include noise measurements inside the ship, which will provide correct noise exposure data to the workers and compare them with the noise perceived. It is also necessary to investigate the proportion of noise pollution topics in maritime education programs, aiming to increase the practical knowledge level and awareness of the noise impacts on health and society.

## Figures and Tables

**Figure 1 ijerph-18-06671-f001:**
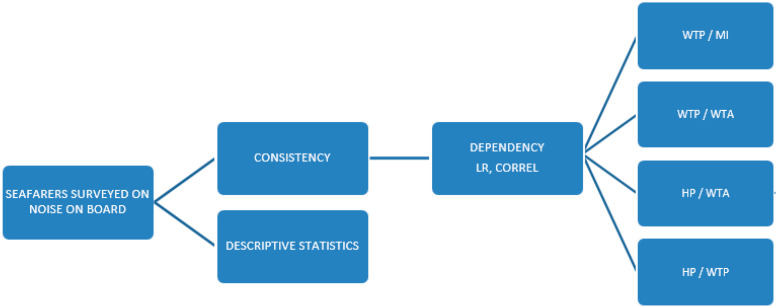
Flow chart of the research.

**Figure 2 ijerph-18-06671-f002:**
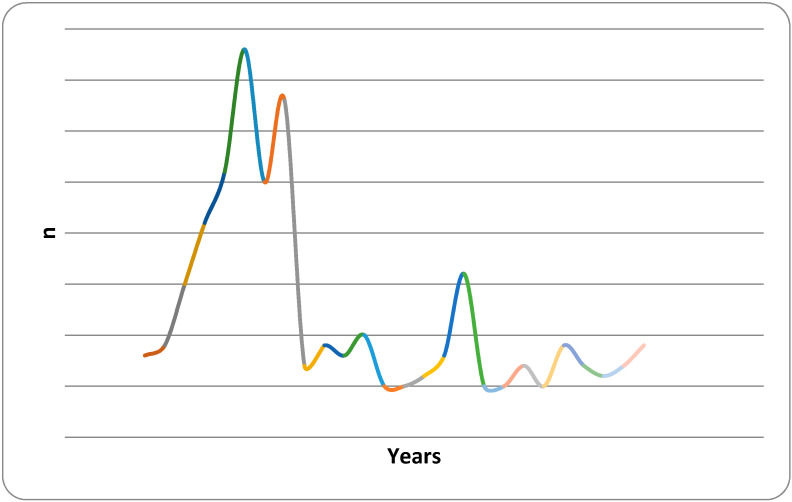
Dispersion of respondents by the work experience.

**Figure 3 ijerph-18-06671-f003:**
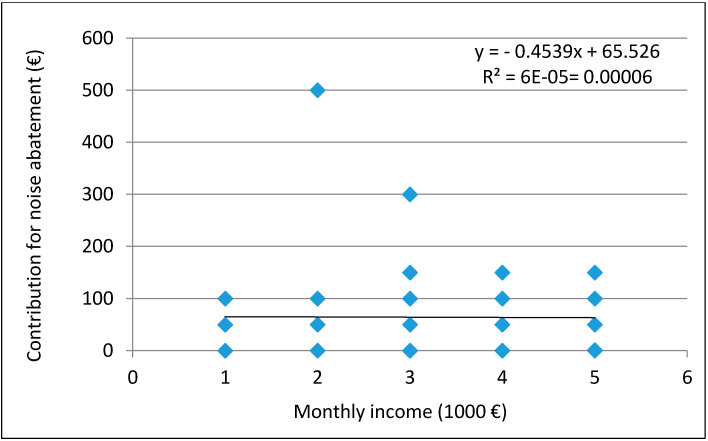
Dependence of declared contributions for noise abatement on monthly incomes.

**Figure 4 ijerph-18-06671-f004:**
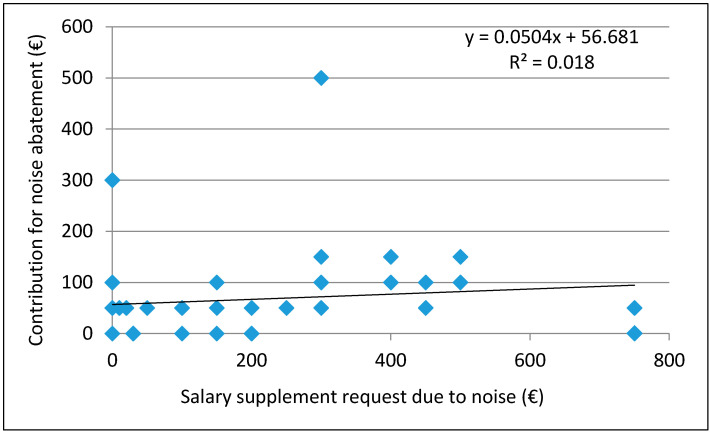
Dependence of annual contribution amount for noise abatement on requests for salary supplement due to noise.

**Figure 5 ijerph-18-06671-f005:**
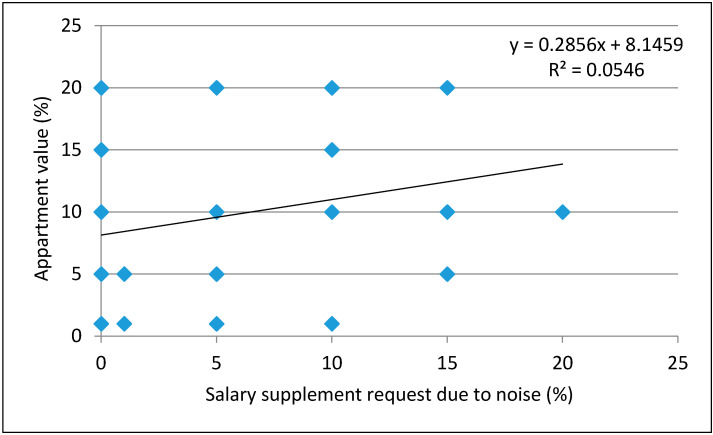
Dependence of attitudes about the noise influence on own apartment value on the amount of request for salary supplement due to noise.

**Figure 6 ijerph-18-06671-f006:**
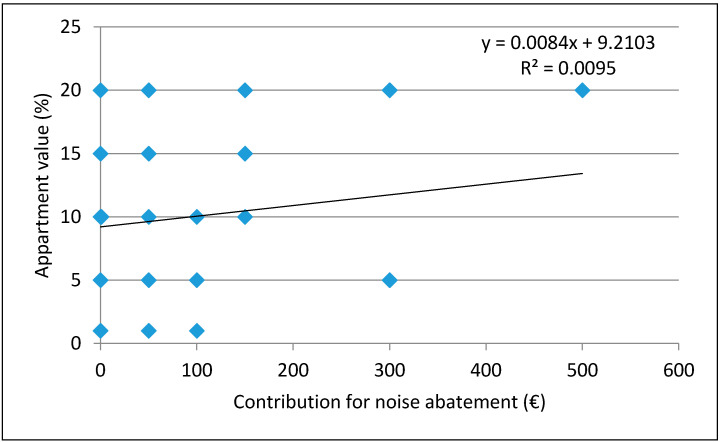
Dependence of attitudes about the noise influence on own apartment value on the declared contribution for noise abatement amount.

**Table 1 ijerph-18-06671-t001:** Perception of the harmful effects of noise on the environment and health.

Perception	Environment		Health		
	pos	neg	pos	pos/neg	neg
General	58.73	41.27	53.97	26.98	19.05
Experience < 10 y	52.17	47.83	43.48	32.61	23.91
Experience ≥ 10 y	76.47	23.53	82.35	11.76	5.88
Secondary school	60.38	39.62	47.17	30.19	22.64
Bachelor/Master	50.00	50.00	90.00	10.00	0.00
Income (1.2)	61.11	38.89	50.00	25.00	25.00
Income (3)	61.90	38.10	40.00	40.00	20.00
Income (4.5)	57.14	42.86	70.00	15.00	15.00

## Data Availability

The data presented in this study are available on request from the corresponding author.

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
