# Peer review of "Seafarers’ Perception and Attitudes towards Noise Emission on Board Ships"

_ijerph, 2021, doi:10.3390/ijerph18126671_

Round 1
Reviewer 1 Report
Overview:
The article presents a study on seafarers’ perception and attitudes towards noise emission on board ships. The study is based on the results of a questionnaire given to 189 subjects (seafarers) in 2020. Statistical analysis showed some interesting results. This research is clearly written. Some comments follow:
Abstract
Abstract should probably include additional information of the results (more details in the results section).
The authors should probably present their findings according to their significance in order to improve the impact of this research (e.g. The most important findings of the research were..).
Introduction
Line 36: ‘noise external costs unit prices’. Can you elaborate more on this?
Line 41: ‘in the transport impact on’. Should be ‘of the transport impact on’.
Line 79: ‘IMO’. International Maritime Organization should be included before the abbreviation.
Line 81: ‘The Code’ (?). The authors should elaborate more on this.
Line 83: ‘..been raised since that’. Do the authors mean ‘..been raised since then’?
Lines 89-110: It seems that this paragraph is better suited for the ‘Materials and Methods’ section (at least lines 99-110). ‘Materials and Methods’ section is more appropriate to explain why you chose the specific methodology (than the introduction).
Materials and Methods
If the authors add the suggested paragraph from the introduction (lines 89-110), it is preferred to include more information (related to noise and acoustics) about WTP and WTA as many readers will not be familiar with them.
Results
Table 1: By ‘High school’, do you mean Bachelor/Master (according to the questionnaire)?
Line 158: The authors should add the specific time period in which the questionnaires were applied.
Line 159: The authors must add where the study took place (where the subjects were found) and how the subjects were chosen.
Line 169: ‘The negative perception among respondents with > 10 years of work experience is significantly lower (23.53%)’. This is an interesting result and the authors should consider including it in the abstract (the percentage).
Line 175: ‘A significantly higher positive perception was recorded in groups of respondents with a university degree (90%), work experience longer than ten years (82.35%), and monthly income higher than € 4,000 (70%)’. Again the authors should consider including the percentages it in the abstract.
Discussion and conclusions
Line 270: ‘The results show that noise pollution on board is not only temporary but permanently’. Permanently should be permanent.
The authors should include a table of abbreviations at the end of the manuscript.
Reviewer 2 Report
This work presents a short analysis regarding to the seafarers' perception and attitudes towards noise emission on board ships.
It is interesting but it only focuses on the noises as a problem to be solved. Maybe the authors need to consider that sometimes, the noises are inevitable because it depends of the machines in use and natural noises of the environment, for instance.
The work also says that "Their perception of the impact of noise on health grows with work experience.." What is the basement of this assertion? Any references? It is interesting to analyse better the data according to the work experience. The plots presented in Section 3 are not clear enough.
Table 1, shows the data according to the work experience too, but it seems to be inaccurate. Separate the work experience on more then 10 and less then 10 years makes no sense actually. The analysis coming from the difference between exp.>10 and the exp.<10 is insignificant of even wrong. How to assume that the data from 11 yrs will differ of 11 yrs? I thing that between the participants, several dispersion plots must to be added before the correlation, to see if makes sense the dataset in use. Also the type of participant, merchant or passengers can be analysed in different perspectives. Passengers (tourist for example), see the situation more like elation or even joy then critical perspective, and it will attenuate the results.
In resume.
We understand the point of the work contribution but unhappiness it seems to be a very simple work to this ERPH-Journal.
The dataset must to be improved regarding to the participant, because the results seems to be inconclusive. Percentage looks so similar in general.
40%/41.27% of the participants don't consider noise to be a significant problem shows that at most 60% consider to be significant problem or maybe. In wild, actually, less then 60% will consider the noise a problem, because some of them, can consider it irrelevant, what it is not a good demand to support a Journal study in this case.
I believe that in this time, the present work is not sufficient to be accepted mainly because the data and results seems to be inconclusive, maybe because the dataset is not well defined. Some plots must to be added using maybe unsupervised analyses, what it will be a plus.
Best regards,
Reviewer 3 Report
Review of the manuscript entitled ‘ Seafarers’ perception and attitudes towards noise emission on board ships
The study deals with ship noise onboard that can endanger seafarers and passengers.
The study tries to determine the seafarers' noise pollution perception on board and evaluate their attitudes towards noise exposure. Author claims that awareness level about the harmfulness of noise in people who are professionally exposed to it and therefore may suffer health consequences is a good indicator of how much significance is attached to noise as an environmental pollutant.
The study is very interesting and deserves to be published. It could impact the working conditions of seafarers.
- However, the study doesn’t precise the Nationality of the seafarers. Does the nationality could have an influence on the survey?
- The negative perception among respondents with > 10 years of work experience is 169 significantly lower (23.53%) compared to respondents with <10 years of work experience 170 (47.83%).
This assertion supposes that with the time the seafarers are accustomed to the noise or have a limited perception of the noise. An interesting study would be to ask to the seafarers to use a smartphone app to measure the noise in the working environment. That measurement would make aware the seafarers of the noise.
Monthly income does not affect the perception of noise pollution. This fact is curious. It supposes that a good salary could limit any perception of the noise.
Health impact perception additionally rises with higher education and incomes. This is true ‘to educate is to form judgment’.
Round 2
Reviewer 2 Report
Dear authors, thank you for you feedback. The work is in progress.
Some points weren't answered yet.
Regarding to Point 1. Each reviewer can't consider the opinion of other reviewer. This way, I continue to follow my previous point because it wasn't improved in the text. Unhappiness, the dataset is not sufficient defined, considering the relevance of the proposal.
The answer for point 5 says " Considering the noise on board as irrelevant is not the option for us. " Why considers only the others. What if 90% of the participants consider it irrelevant? What we should do? Consider only the 10%? It seems incorrect. The research must to considers all perspectives in case, and after that the authors can narrow this answers but linking the differences.
If the authors say "it is firstly important to protect the participants who are permanently on board" so, the irrelevant answer must to be present even between the participants permanently on board, why not? What if only one participant says to have problem with the noise? The problem is the noise indeed, or he has other problems? To considers the irrelevant answer makes to research better in analysis context.
My prior questions about Table 1, weren't answered. I thing the authors forgot it.
Since some important answers weren't executed as recommended, I decided to wait for major reviews.
